# Can We Use *Ginkgo biloba* Extract to Treat Alzheimer’s Disease? Lessons from Preclinical and Clinical Studies

**DOI:** 10.3390/cells11030479

**Published:** 2022-01-29

**Authors:** Liming Xie, Qi Zhu, Jiahong Lu

**Affiliations:** State Key Laboratory of Quality Research in Chinese Medicine, Institute of Chinese Medical Sciences, University of Macau, Macau SAR 999078, China; mb95810@umac.mo (L.X.); yb67536@um.edu.mo (Q.Z.)

**Keywords:** *Ginkgo biloba* extract, Alzheimer’s disease, clinical trial, meta-analysis

## Abstract

(1) Background: *Ginkgo biloba* extract (GBE) has been widely used to treat central nervous system and cardiovascular diseases. Accumulating evidence has revealed the therapeutic potential of GBE against Alzheimer's disease (AD); however, no systematic evaluation has been performed; (2) Methods: a total of 17 preclinical studies and 20 clinical trials assessing the therapeutic effects of GBE against AD were identified from electronic databases. The data in the reports were extracted to conduct a meta-analysis of the AD-related pathological features or symptoms; (3) Results: For the preclinical reports, 45 animals treated with GBE, in six studies, were subjected to cognitive function assessments by the Morris water maze. GBE was shown to reduce the escape latencies in several studies, in both rats and mice (I^2^ > 70%, *p* < 0.005). For the clinical trials, eight trials, including 2100 individuals, were conducted. The results show that GBE improved the SKT and ADAS-Cog scores in early-stage AD patients after high doses and long-term administration; (4) Conclusions: GBE displayed generally consistent anti-AD effects in animal experiments, and it might improve AD symptoms in early-stage AD patients after high doses and long-term administration. A lack of sample size calculations and the poor quality of the methods are two obvious limitations of the studies. Nevertheless, the preclinical and clinical data suggest that further large-scale clinical trials may be needed in order to examine the effects of long-term GEB administration on early-stage AD.

## 1. Introduction

Alzheimer’s disease (AD) is an irreversible, age-related, and progressive neurodegenerative disease [1]. In 2019, Alzheimer’s Disease International (ADI) estimated that more than 50 million people suffer from dementia worldwide, and they predict that the number will rise to 152 million by 2050, which would place an increasingly severe burden on societies and national economies [2]. At the same time, there are no known effective treatments for AD. Finding an effective therapy for AD would help millions of people, both directly, by improving their health, and indirectly, by relieving the burden on healthcare systems.

In terms of pathogenesis, amyloid plaques and neurofibrillary tangles are the two major structural changes in AD brains [3]. Currently, the amyloid hypothesis and the tau hypothesis are the generally accepted theories used to explain the etiology of AD. Amyloid-β (Aβ) is the amyloid-β precursor protein (AβPP) cleavage product of a membrane protease called “secretase”, and Aβ aggregation is the major component of amyloid plaques in AD patients. Hyperphosphorylated tau aggregation is another major reason for the neurofibrillary tangles in AD. At present, the drugs on the market only alleviate the clinical symptoms, and cannot reverse the pathological changes and disease course of AD [4,5]. On the basis of the current research, there is an urgent need to seek out a disease-modifying agent to slow down the progression of AD.

*Ginkgo biloba* is a famous herbal medicine that has been used in China since ancient times. In the 1970s, a standardized *ginkgo biloba* leaf extract (GBE), containing multiple pharmacologically active substances, was developed by Dr. Willmar Schwabe (Karlsruhe, Germany) [6]. It is a dry extract from *G. biloba* leaves (35–67:1), and the extraction solvent is acetone (60% *w*/*w*). The extract contains 22.0–27.0% ginkgo flavonoids, including quercetin, kaempferol, and isorhamnetin [7], 5.0–7.0% terpene lactones, consisting of 2.8–3.4% ginkgolides A, B, and C and 2.6–3.2% bilobalide, and ginkgolic acids at levels less than 5 PPM [8]. The main ingredients of standardized *gingko biloba* extract are flavonoids and terpenoids [9,10]. These components may be responsible for GBE’s effects in the treatment of AD, which include: antioxidation, anti-inflammation, and antiapoptosis; protection against mitochondrial dysfunction, amyloidogenesis, and Aβ aggregation; the modulation of the ion homeostasis and phosphorylation of the tau protein; and even the induction of growth factors [11]. Several studies report that GBE significantly improved the performances of AD mice in the Morris water maze test [12]. Since 1985, a number of clinical trials have been conducted to evaluate the anti-AD efficacy of GBE. However, limited sample sizes and methodology flaws cast doubt on the value, accuracy, and reliability of these results. A systematic review is urgently needed to clarify what science has actually learned about the efficacy of GBE in treating AD.

This review reports the results of our systematic meta-analysis of articles on GBE for AD treatment in preclinical and clinical studies. In addition, we summarize the potential neuroprotective mechanisms of GBE in AD, determined from animal models. More importantly, we analyze and summarize the possible causes of the inconsistencies by comparing the effective and ineffective clinical trials. This review also provides a reference for the assessment of the methodological quality of AD preclinical and clinical studies.

## 2. Materials and Methods

### 2.1. Search Strategy

First, to identify articles focusing on the efficacy of *Gingko biloba* extract on AD animal models, a careful search was performed of the literature published between 2000 and 2020 and found in the electronic databases, Web of Science and PubMed. The key words used for the article search were (“Alzheimer”) AND (“*Ginkgo Biloba*”). We also conducted a search with Google Scholar and included the first 200 hits, sorted by relevance [13]. Two individual reviewers (Liming Xie and Qi Zhu) worked independently to screen the abstracts of the qualified articles on the basis of the inclusion criteria (Table 1). When we disagreed, the article was screened by a third reviewer (Erjin Wang). Similarly, the clinical trial results were searched with three key words, (“Alzheimer disease”) OR (“Dementia”) AND (“Gingko biloba”), in the ClinicalTrials.gov website, and were searched with four key words, (“clinical trial”), (“Alzheimer disease”) OR (“Dementia”) AND (“Gingko biloba”), in Google Scholar.

### 2.2. Selection Criteria

The inclusion and exclusion criteria for the selection of preclinical trials are listed in Table 1, and the criteria for the selection of clinical trials are presented in Table 2.

### 2.3. Data Extraction and Analysis

After finishing the screening, two individual researchers extracted and tabulated the data from the selected articles. The details of the preclinical articles are listed in Table 3, with the following items: (1) First author name and publication year; (2) AD animal model; (3) Strains, weights, and sex of animals used; (4) Treatment dosage, duration, and administration route; and (5) Methods used in the experiments, with results corresponding to the methods. Specific information for the clinical study articles is listed in Table 4, with the following items: (1) Study, author(s), publication date; (2) Country; (3) Inclusion criteria; (4) Setting of study; (5) Duration; (6) Treatment; (7) Groups; (8) Age of participants; (9) Baseline scale; and (10) Withdrawal rate. Image J software was used to extract the numerical values from the graphs.

A meta-analysis was conducted with Review Manager 5.3 software. We processed the data with a fixed effects model and judged the heterogeneity with a Q test and I^2^ statistics. Heterogeneity was considered to exist when *p* < 0.05: I^2^ = 0% means no heterogeneity; 0 < I^2^ ≤ 25% means mild heterogeneity; 25% < I^2^ ≤ 75% means moderate heterogeneity; and I^2^ > 75% means a high degree of heterogeneity.

### 2.4. Quality Assessment

We scored the methodological quality of the included articles using the CAMARADES (Collaborative Approach to Meta-Analysis and Review of Animal Data from Experimental Stroke) list. Additionally, the preclinical study evaluation criteria were established in accordance with the characteristics of Alzheimer’s disease. The criteria were as follows: (1) Publication in a peer-reviewed journal; (2) Random allocation of animals; (3) Outcomes assessed blindly; (4) Dose–response relationship assessed; (5) Appropriate animal model; (6) Necessary sample size calculation; (7) Observation of animal welfare regulations; and (8) No potential conflicts of interests [49]. Each article was given a quality score out of a maximum total of eight points. The clinical studies were evaluated similarly, with the addition of one more criterion: an ITT analysis (intent-to-treat analysis).

## 3. Results

### 3.1. Study Selection

#### 3.1.1. Screening of Preclinical Studies

The basic search produced a total of 1038 articles from the electronic databases, Web of Science, PubMed, and Google Scholar. After browsing the titles and abstracts of the articles, we excluded 200 articles according to the following criteria: (1) Review articles, comments, letters, case reports (*n* = 305); (2) Clinical trials (*n* = 273); and (3) The efficiency of the GBE was tested on AD cell models (*n* = 43). Next, by carefully screening the full texts of the remaining 292 articles, 275 articles were excluded for the following four reasons: (1) The GBE was tested on nonmammalian animal models (zebra fish, Caenorhabditis elegans, drosophila, etc.); (2) The GBE was not tested on an AD model; (3) Other drugs were used in combination with GBE; and (4) The full text was missing. Finally, we included 17 articles. The details are shown in Figure 1.

#### 3.1.2. Recruitment Status of GBE Clinical Trials

As is shown in the Figure 2, a total of 98 clinical trials related to *G. biloba* were identified in the ClinicalTrials.gov website or in Google Scholar. More than half of those clinical trials have been completed (*n* = 57). The statuses of the remainder were classified as: unknown (*n* = 13); active, not recruiting (*n* = 2); terminated (*n* = 3); withdrawn (*n* = 3); enrolling by invitation (*n* = 1); recruiting (*n* = 14); and not yet recruiting (*n* = 5). In this systematic review, 20 clinical-trial-related articles have been included in order to conduct a further meta-analysis and a methodological quality assessment.

### 3.2. Article Characteristics

#### 3.2.1. Analysis of Included Preclinical Studies

In the preclinical studies reported in the 17 articles, two animal species were used: mice (*n* = 6), and rats (*n* = 11). Four strains of mice were used: C57BL/6 mice (*n* = 1) [12], Tg2576 mice (*n* = 1) [14], TgAPP/PS1 mice (*n* = 3) [18,26,50], and TgCRND8 APP-transgenic mice (*n* = 1) [25]. Two strains of rats were used: Wistar rats (*n* = 5) [15,16,17,19,23], and Sprague–Dawley rats (*n* = 6) [21,22,24,27,28,29]. In twelve studies, only male animals were used, while, in four studies, only female animals were used; one article did not mention the sex of the animals. The animal numbers (from 8 to 36) and the animal ages varied greatly among the articles. The details are shown in Table 3.

Half of the studies established their AD models by using drugs, such as AlCl_3_ (*n* = 4) [15,17,28,29], scopolamine (*n* = 1) [23], hyperhomocysteinemia (*n* = 1) [27], and Aβ^25–35^ (*n* = 3) [21,22,24]. These toxin-induced AD models have a common feature of mimicking the pathological alterations and cognitive impairment of AD, but each has its drawbacks. Specifically, AlCl_3_ (aluminum) has a significant effect on the enzyme activity, which influences protein synthesis and the neurotransmitter activity, and moreover, this method spends a long time on modeling. Scopolamine is a well-known M-cholinergic receptor blocker, which can damage the cholinergic neuron. However, scopolamine-treated animals lack the typical pathological features of AD; that is, scopolamine fails to induce the irreversible nerve damage in AD. Aβ injection causes an accumulation of Aβ, leading to plaque formation and neuron toxicity. However, this acute toxicity model cannot reflect the relatively slow neurodegeneration process of AD. Hyperhomocysteinemia seems applicable for AD modeling, and can significantly increase the plasma Hcy levels for memory impairment and tau hyperphosphorylation in rats.

Transgenic mice were used as AD models in five studies (29.4%), and wild-type naturally aging animals were used in the remaining three studies (17.6%). Transgenic mice, including Tg2576 mice, TgAPP/PS1 mice, and TgCRND8 APP-transgenic mice, are suitable to represent AD pathogenesis, and have been widely used for pharmacological testing in preclinical studies. In some studies, researchers chose the natural aging model, with the characteristics of late onset in actual AD patients. This model can better reflect the therapeutic effects of drugs, but it fails to simulate the key pathological manifestations of AD, including amyloid plaques and NFT.

#### 3.2.2. Analysis of Included Clinical Studies

A total of 14 of the 20 included clinical studies (70.0%) conclude that GBE can effectively improve the cognitive ability of AD patients, while others report no significant improvement. The criteria used in the 20 clinical studies were: DSM-III-R criteria [51]; ICD-10 criteria [52]; the Mini-Mental State Examination (MMSE) [53]; the Alzheimer’s Disease Assessment Scale Cognitive Subscale (ADAS-Cog) [54]; the Global Deterioration Scale score, NINCDS-ADRDA criteria [55]; and the Syndom Kurztest (SKT) [20]. The baseline scales of each clinical trial are listed in Table 4, including the cognition conditions (SKT/MMSE/ADAS-Cog), ages, and sex ratios. Among the 14 effective studies, 7 (50.0%) chose a GBE dose of 240 mg per day [8,33,34,36,38,39,42]; 4 (28.6%) chose a dose of 120 mg/day [31,32,35,41]; 1 selected 160 mg/day [37], and 1 selected 80 mg/day [30]. The dose used in the remaining study was unclear [40]. In the included effective studies, the duration of the drug administration ranged from 2 weeks to 20 years. Interestingly, 10 studies (71.4%) chose a relatively long drug administration period (over 20 weeks) [8,32,33,35,36,37,38,39,41,42]. A total of two clinical studies tracked 12 weeks [31,34], one clinical study lasted for 20 years [40], and the other one lasted for only 2 weeks [30]. According to the raw data from the clinical trials, the comparison of the GBE effects on AD symptoms was made according to the age groups (Figure. 3A). The age groups of AD patients that showed effective outcomes were mainly distributed in the 60–70-year-old (*n* = 544) and 70–80-year-old (*n* = 588) groups. Most of the AD patients who showed no significant improvement for years after GBE treatment were over the age of 70 (70–80: *n* = 2116; 80–90: *n* = 1072) (Figure 3A). Actually, when we calculate the effective ratio among the different age groups, we find an obvious decline with increasing age, indicating that GBE may be more effective in younger populations, whose AD-related damage is normally mild. The details are shown in Figure 3B.

In the studies that reported negative outcomes, only two of six studies were conducted over periods of five years [46,47], and the majority of the patients in these two studies were over 75 years old. We speculate that, in the elderly population, the disease has progressed to an irreversible condition that cannot be alleviated by GBE administration. Another failed clinical trial lasted for only six weeks [56]. Schneider et al. (2005) [44] conducted a randomized placebo-controlled double-blind trial with GBE administration for 26 weeks. However, when they found a lack of cognitive decline in patients taking a placebo, they suspected that the assessment criteria may not have been sensitive enough to detect a treatment effect, and they considered the results inconclusive. Similarly, McCarney et al. (2008) [45] and Nasab et al. (2012) [48] conducted trials with daily administrations of 120 mg GBE for 24 weeks. Nasab compared the efficacy of GBE with that of rivastigmine. Both trials confirmed the safety of GBE, but Nasab considered that rivastigmine, a representative cholinesterase inhibitor, performed better than GBE in AD treatment. Nonetheless, the result of the clinical trial was considered insignificant. All three trials had subpositive results with ambiguous conclusions. The details are shown in Table 4.

### 3.3. Meta-Analysis Results

#### 3.3.1. Behavioral Test Analysis in Preclinical Studies

The Morris water maze (MWM) test is a classical experiment used to evaluate the cognitive functions of learning and memory in AD animal models [57]. The time of crossing to the platform indicates the spatial memory ability, in terms of the memory storage and extraction capability, while the escape latency in the spatial probe test is considered to be an indicator of the spatial learning ability.

A total of 4 of the 17 (23.5%) articles detected the numbers of times crossing the target quadrant in the MWM to evaluate the spatial memory in the experimental AD animals. A total of 42 animals (rats and mice, *n* = 32 and *n* = 10, respectively) were included and treated with different doses of GBE (100 mg/kg in one article, 50 mg/kg in one article, 40 mg/kg in one article, and 20 mg/kg in one article). A total of 42 animals (rats and mice, *n* = 32 and *n* = 10, respectively) were treated as control groups. The univariate statistical analysis was conducted using Revman 5.3 software. In the experimental AD model, whether it was mice or rats, GBE significantly increased the numbers of times animals crossed the target quadrant, when compared with vehicle-treated animals (*p* < 0.00001), with a Std. mean difference of 2.79. Notably, the results of the forest plot show moderate heterogeneity (*p* = 0.02, I^2^ = 69%), which indicates that GBE has a potential relationship with the increased crossing times of the AD animal model. The details are shown in Figure 4.

A total of 6 of the 17 (35%) articles tested the escape latency in the spatial probe test of the MWM to assess the spatial learning ability in the experimental AD animals. A total of 45 animals (rats, *n* = 31, mice, *n* = 14, respectively) were included and treated with different doses of GBE (100 mg/kg in two articles, 400 mg/kg in one article, 60 mg/kg in one article, and 50 mg/kg in two articles). The univariate statistical analysis was conducted using Revman 5.3 software. In a forest plot, GBE showed a strong curative effect in decreasing the escape latency (*p* < 0.00001), compared with vehicle treatment in the MWM, for both mice and rats, with a mean difference (MD) between 30.54 and 40.02. Moreover, the subgroup outcomes of the forest plot show no heterogeneity (I^2^ = 0%, *p* = 0.56, I^2^ =0, *p* = 0.92). The results of the combination analysis indicate that GBE improved behavior and cognitive impairment (*p* < 0.00001), with moderate heterogeneity (I^2^ = 74%, *p* = 0.001). Differences in the drug dosages, the methods of measurement, and the strains of animals are possible reasons for the higher heterogeneity. The details are shown in Figure 5.

#### 3.3.2. Cognition Improvement Analysis in Clinical Studies

The Syndom Kurztest (SKT) and the Alzheimer’s Disease Assessment Scale Cognitive Subscale (ADAS-Cog) are two widely used cognitive tests that can accurately reflect the attention and memory deficits of AD patients [20,54]. The cognition tests, such as SKT and ADAS-Cog, were evaluated at the beginnings and the ends of the clinical trials. A total of 7 of 10 (70%) effective studies were included in the meta-analysis of AD patients in the SKT. A total of 819 patients who had been diagnosed with AD, or that had AD-like symptoms, received lower SKT scores after GBE treatment for 12 to 24 weeks. The univariate statistical analysis was conducted using Revman 5.3 software. In the forest plot, the changes in the SKT scores ranged from −3.2 to −0.0075 in the GBE treatment group, compared with −1.2 to 1.3 in the placebo group, with SMD (95%CI) = −2.44 [−2.60, −2.29]. Although the study results show that GBE can ameliorate cognitive deficits, the forest plot shows that high heterogeneity (*p* < 0.00001, I^2^ = 99%), different drug dosages, and different administration times are possible reasons for the differences between the experimental and placebo groups. The details are shown in Figure 6.

Next, the subgroup analysis of the SKT by dose included six effective studies. A total of 779 patients diagnosed with AD or cognitive impairment were treated with GBE (daily dose below 240 mg, *n* = 104; daily dose of 240 mg, *n* = 675), while 703 patients were treated with placebos (daily dose below 240 mg, *n* = 70; daily dose of 240 mg, *n* = 633). The univariate statistical analysis was conducted using Revman 5.3 software. In the forest plot, the changes in the SKT scores ranged from −3.3 to −0.7 in the GBE groups at doses below 240 mg/day, and from −1.2 to 0.95 in the placebo groups, with the SMD (95%CI) = −0.38 [−0.71, −0.05]. At the dose of 240 mg/day, the changes in the SKT scores ranged from −2.225 to −1 in the GBE groups, and from −1.2 to 2.4 in the placebo groups, with the SMD (95%CI) = −0.59 [−0.71, −0.48]. These results indicate that treating with 240 mg/day of GBE is more effective than treating with 160 mg/day. The forest plot results show high heterogeneity (*p <* 0.00001, I^2^ = 97%) in both subgroups of daily doses below 240 mg, and of daily doses of 240 mg. Differences in the drug administration times, the methods of measurement, and individual variations were the potential reasons. The details are shown in Figure 7.

A total of 4 of the 10 (40%) effective studies evaluated the cognitive ability with ADAS-Cog. A total of 559 patients diagnosed with AD or cognitive impairment were treated with GBE (daily dose below 240 mg, *n* = 380; daily dose of 240 mg, *n* = 179), while 571 patients were treated with placebos (daily dose below 240 mg, *n* = 388; daily dose of 240 mg, *n* = 183). The univariate statistical analysis was conducted using Revman 5.3 software. The meta-analysis shows that GBE decreased the ADAS-Cog score from 31.21 to 30.33, with the SMD (95% CI) = −0.64 [−0.78, −0.51]. Moreover, in the forest plot of the subgroup analysis, the changes in the ADAS-Cog scores for patients taking less than a 240-mg daily dose ranged from −0.8 to −0.2 in the GBE treatment groups, and from 1.3 to 2.8 in the placebo groups, with the SMD (95% CI) = −1.52 [−1.69, −1.35]. At the dose of 240 mg/d, the changes in the ADAS-Cog score ranged from −0.88 to −0.6 in the GBE treatment groups, and from 0.03 to 2.8 in the placebo groups, with the SMD (95% CI) = −0.59 [−0.80, −0.38]. According to these results, GBE also can decrease the ADAS-Cog scores of AD patients. The details are shown in Figure 8.

### 3.4. Neuroprotective Mechanism Analysis

After screening the full texts of the selected articles, we found five mechanisms that have been suggested for the neuroprotective mechanisms of GBE in AD models: reducing Aβ deposits and p-Tau; anti-oxidation; antiapoptosis; anti-inflammation; and neurotransmitter balance regulation.

#### 3.4.1. GBE Can Significantly Reduce Aβ Deposits and p-Tau in AD Models

Previous studies have confirmed that neuronal plaques (SP), formed by amyloid-β deposits (Aβ), and neurofibrillary tangles (NFT), caused by hyperphosphorylated tau aggregation, impair the learning and memory functions of the brain (Figure 9). As a result, the inhibition of Aβ aggregation and tau hyperphosphorylation have been two major strategies for alleviating AD symptoms [3].

A total of 3 of the 17 (17.6%) articles detected the Aβ levels in the brain tissues with immunoblotting. A total of ten TgAPP/PS1 mice were included and treated with different doses of GBE (100 mg/kg in one article, 50 mg/kg in two articles,). Another ten TgAPP/PS1 mice were treated as a control group. The univariate statistical analysis was conducted with Revman 5.3 software. The forest plot results indicate that GBE lessened the Aβ deposits, both in the hippocampus and the cortex, compared with vehicle-treated animals (*p <* 0.00001), with a mean difference of 43.45. Notably, the results of the forest plot show moderate heterogeneity (*p* = 0.11, I^2^ = 54%), indicating that GBE contributed to the reductions in the Aβ deposits. The results of the Aβ deposits were consistent with those of the behavioral test analysis. The details are shown in Figure 10.

Zeng et al. (2018) [27] discovered that GBE can ameliorate the HHcy-induced tau hyperphosphorylation in the hippocampus and prefrontal cortex, and that it can further reduce the dendritic and synaptic plasticity to improve neurodegeneration. Verma et al. (2020) [29], using staining methods in an AlCl_3_-induced AD model, made it evident that GBE administration can decrease the expression of phosphorylated tau protein.

#### 3.4.2. GBE Displays the Antioxidant Activity

Much research has confirmed that oxidative stress damage is a pathological feature of early AD. In the process of brain degradation, unsaturated fatty acids in the cell membranes of neurons are continuously oxidized, producing a large number of free radicals and generating peroxide lipids (LPOs). Among them, malondialdehyde (MDA) is the most toxic metabolite, and it interferes with the normal metabolism and function of neurons. The human body has free radical scavenging systems, including superoxide dismutase (SOD), glutathione peroxidase (GSH-Px), and catalase (CAT). Hence, inhibiting oxidative stress and boosting the scavenging of free radicals should be effective methods for treating AD (Figure 9).

Six articles have demonstrated that GBE has an antioxidative effect. Tian et al. (2012) [21] and Tian et al. (2013) [22] report that the SOD and GSH levels were significantly increased in the hippocampi of Aβ-injection-induced AD model animals, while a significant reduction was detected in the content of MDA after GBE treatment. Protein carbonyls are often a marker of oxidative damage to proteins. Stackman et al. (2003) [26] discovered a significant increase in such carbonyls in Tg2576 mice chronically treated with GBE. In this case, because the GBE improved spatial learning, the increase in carbonyls may reflect a change in the brain metabolism, and not oxidative damage. Zeng et al. (2018) [27] and Verma et al. (2020) [29] report that GBE had protective effects through reducing the levels of oxygen-derived free radicals/reactive oxygen species (ROS) in hyperhomocysteinemia-induced and AlCl_3_-induced models. Furthermore, Verma et al. (2019) [28] confirm that GBE enhanced the GSH levels and GPx activity, while it decreased the GSSG levels and GST activity, in both the hippocampus and the cortex.

#### 3.4.3. GBE Inhibits Cell Apoptosis

Multiple studies have confirmed that extensive neuronal loss is one of the major pathological features of AD, leading to learning and memory impairment in AD patients. The apoptosis mechanism of AD is closely related to: the aggregation of Aβ; the downregulation of apoptosis protein bcl-2 expression; the activation of the proapoptotic protein, Bax; and the activation of caspase. As a consequence, antagonistic neuron apoptosis has become an important topic in the intervention of AD. The details are shown in Figure 9.

A total of 4 of the 16 (25%) articles show that GBE can inhibit cell apoptosis. The activities of caspase-3 and caspase-9 play important roles in cell apoptosis. Gong et al. (2005) [15] and Tian et al. (2013) [22] discovered that GBE decreases the levels of caspase-3, caspase-9, and Bax to suppress apoptosis in AlCl_3_-induced and Aβ-induced AD models. In addition, both Jahanshahi et al. (2013) [23] and Zhang et al. (2015) [24] directly observed apoptotic cells using TUNEL staining. The images confirm that GBE protects neurons against apoptosis in the hippocampi of scopolamine-induced and Aβ-injection rat models.

#### 3.4.4. GBE Has Anti-Inflammatory Activity

The brain’s inflammatory response is an important feature of AD, and it causes a large number of neurons to undergo apoptosis in AD patients. The activation of microglial cells gives rise to the neuroinflammatory response. Aβ can activate astrocytes and microglia, and can subsequently activate many inflammatory mediators, such as inflammatory cytokines (the typical inflammatory cytokines are IL-1β, IL-6, and TNF-α), nuclear transcription factors (NF-κB), and chemokines (Figure 9). These inflammatory mediators directly damage neurons in the brain and induce the excessive aggregation of Aβ, initiating a vicious cycle of continuous escalating damage.

In this review, five articles report that GBE had anti-inflammatory effects. Hou et al. (2010) [50] found that the oral administration of GBE increased the BDNF levels in the hippocampi of Tg APP/PS1 mice. Consistently, GBE enhanced the levels of phosphorylated CREB, which regulate the expression of BDNF in the hippocampus. Tian et al. (2012) [21] proved the protective effects of GBE by showing that the densities of damaged neurons in an Aβ-induced model decreased. Zhang et al. (2015) [24] reveal that GBE can activate the NF-κB pathway to ameliorate the inflammatory response. Moreover, the levels of NF-κB-related proteins, p-IKKα/β, p-IκBα and p-NFκB, were significantly increased in the Aβ-induced model. Liu et al. (2015) [25] discovered that GBE can markedly downregulate the transcription levels of both proinflammatory and anti-inflammatory (TNF-α, IL-1β, CCL-2, and IL-10), which indicated neurotoxic inflammatory inhibition in the TgCRND8 APP-transgenic mice. The evidence indicates that GBE might affect the phenotype of microglial cells. Wan et al. (2016) [26] found that treatment with GBE downregulated the proinflammatory mediators, and upregulated the anti-inflammatory cytokines, in APP/PS1 mouse brains.

#### 3.4.5. GBE Significantly Improves the Choline System

The loss of cholinergic neurons is recognized as one of the major causes of dementia in the course of AD. Cholinergic neurons are severely damaged in the brains of AD patients. At the same time, choline acetyltransferase (ChAT) activity significantly decreases, while acetylcholinesterase (AChE) activity is enhanced in the brain tissue. Therefore, the level of acetylcholine is downregulated, and finally leads to dementia symptoms, including memory decline and cognitive dysfunction. Improving the function of the cholinergic nervous system is another strategic way to ameliorate AD. The details are shown in Figure 9.

K. Blecharz-Klin et al. (2009) [19] detected regional brain monoamine levels to find the correlation between GBE and the regulation of the neurotransmitter balance. The results show that GBE significantly increased the NA in the prefrontal cortex and hippocampus, the DOPAC in the prefrontal cortex, and the 5-HT in the striatum, all of which correlated with improved spatial memory. Increased acetylcholinesterase activity can lead to decreased acetylcholine levels, and then to reduced cognitive levels. Gong, Q. H. et al. (2006) [17], Verma et al. (2019) [28], and Verma et al. (2020) [29] discovered that AChE activity was significantly decreased in the hippocampi and cortexes of the AlCl_3_-induced AD model after GBE administration.

### 3.5. Methodological Quality Analysis

#### 3.5.1. Preclinical Studies

The scores of the methodological quality for the included preclinical studies ranged from 3 to 5, out of a total of 8 points. As is shown in Table 5, all the selected articles were published in peer-reviewed journals and complied with the relevant animal welfare regulations. A total of 11 of 17 (64.7%) [12,15,16,17,19,21,22,23,24,25,29] articles allocated animals to treatment groups or control groups randomly. Among the remaining six studies, only Stackman et al. [14] report that the behavioral tests were conducted blindly, indicating irregularities in the design of the experiments. Therefore, in order to reduce the subjective differences, which result in poor data quality, we suggest that animal experiments, especially the behavior test, should be performed in a blind manner. A total of 8 of 17 (47.1%) [19,21,23,24,26,27,28,29] articles attested that they were free of potential conflicts of interest. Most of the unclaimed studies, including those of Ward et al. [12], Stackman et al. [14], Gong et al. [15], Wang et al. [16], Gong et al. [17], and Tchantchou et al. [18], were published before 2008. The lack of rigorous and complete paper-writing norms in the early stages may be responsible for this phenomenon. Only three (17.6%) [15,16,17] articles assessed the dose–response relationships. Wang et al. [16] allocated Wistar rats randomly to a control group, a 30 mgkg^−1^ EGb761 group, or a 60 mgkg^−1^ EGb761 group, while Gong et al. [15,17] divided Wistar rats randomly to a control group, a 50 mgkg^−1^ EGb761 group, a 100 mgkg^−1^ EGb761 group, or a 200 mgkg^−1^ EGb761 group, after AlCl_3_ treatment. On the one hand, the samples of the animal experiment should be designed to be large enough to achieve reliable outcomes. On the other hand, animal injuries should be reduced as much as possible. A sample size calculation, to achieve sufficient power for the statistical significance, should be conducted; however, it was not performed in any of the included studies. These flaws in the methodologies of the studies obviously affect the reliability of the conclusions. All the details on the methodology qualities are shown in Table 5.

#### 3.5.2. Clinical Studies

The methodological quality scores of the included clinical trials ranged from 3 to 7, out of a total of 9 points. As is shown in Table 6, all of the selected clinical studies were published in peer-reviewed journals. Only two articles did not allocate patients to treatment groups or control groups randomly, or assess the outcomes blindly [40,41]. Amieva et al. [40] conducted an exploratory retrospective analysis of longitudinal data collected prospectively over twenty years of follow-ups of the PAQUID cohort. Similarly, Canevelli et al. [41] evaluated the effects of EGb761 in AD patients receiving cholinesterase inhibitors from a cohort study. Both of the two analyses were from cohort studies, not from randomized controlled trials. Significantly, only Schneider et al. [44] assessed the dose–response relationship of EGb761, at 120~240 mg per day. A total of 16 of 20 (80.0%) [8,30,31,32,33,34,35,38,39,42,43,44,45,46,47,48] articles report the specific reasons for withdrawals during the clinical trials. Kanowski and Hoerr [36] report that a total of 222 patients were enrolled in the clinical trial, but only 205 patients were included in the analysis, with no explanation for the withdrawals. Mazza et al. [37] enrolled 117 AD patients and excluded 41 patients, without specifying the criteria. Canevelli et al. [41] and Amieva et al. [40] conducted cohort studies with no specific descriptions of the withdrawals. A total of 6 of 20 (30.0%) studies conducted ITT analyses and calculated the sample sizes necessary to achieve sufficient power. One-third of the included studies had sample sizes of more than 300, and Amieva et al. [40] and Vellas et al. [47] collected more than 2000 samples. All of the details are shown in Table 6.

## 4. Discussion

### 4.1. Active Components in Gingko biloba Extract with Anti-AD Properties

Alzheimer’s disease is a brain disease with a high and increasing incidence in elderly people, and there is currently neither a cure nor an effective treatment. Currently, the drugs approved by the FDA, such as donepezil, galantamine, tacrine, and memantine, only improve the disease symptoms, without modifying the disease process, and most of them have obvious adverse effects. In general, the occurrence of AD is in combination with many factors, and, when some of these factors fail, new strategies arise. The amyloid hypothesis and the tau hypothesis are two mainstream explanations for the etiology of AD. However, these hypotheses are losing favor as they have failed to yield effective treatments or drugs.

The earliest use of ginkgo in China can be traced back 5000 years [58]. Today, it has been extracted and to isolate a variety of constituents, including bilobalide, ginkgolides A-C, quercetin, isorhamnetin, hydroxykinurenic, rhamnose, glucose, and kaempferol. Bilobalide and ginkgolides A–C are terpenoids. They specifically inhibit platelet-activating factor (PAF) receptors [59], and are considered to be the most promising natural PAF receptor antagonists in clinical application. It is worth noting that ginkgolide B can attenuate the neurotoxicity induced by β-amyloid [10,60,61]. Quercetin, isorhamnetin, and kaempferol are flavonoids [62]. Flavonoids are widely reported for their anti-free-radical and antioxidant effects. In addition, flavonoids have good effects in preventing and treating cardiovascular diseases, such as preventing arteriosclerosis, lowering blood lipid and cholesterol, lowering blood sugar, dilating blood vessels, improving vascular permeability, and reducing the incidence of coronary heart disease [63,64,65]. The administration of flavonoids could be a particularly good strategy for the prevention and treatment of the vascular symptoms related to AD.

### 4.2. Article Characteristics

The choice of an animal model is crucial to the determination of the value of the experimental results obtained. Of the 17 included articles, 9 (52.9%) selected the toxin-induced AD model. They have the common feature of reflecting cognitive impairment. However, acute-toxicity-induced models cannot simulate the process of neurodegeneration, and these models lack NFT formation and Aβ deposition. Currently, transgenic technology is the most advanced and promising measure for establishing AD models. However, only eight of the studies (out of 17, 47%) reviewed here used transgenic-mice AD models (APP/PS1, CRND8, Tg2576) to evaluate the anti-AD effect of GBE. Their mice models showed a rapid progression of AD symptoms, but a lack of tau pathology. Future studies, using more widely accepted transgenic-mice AD models, such as the 3xTg and 5xFAD strains, would be useful for the further evaluation of the therapeutic effects of GBE on AD.

Although the collated data from the clinical studies assessed in this review and the methodologies of the studies were not ideal, the data seem to indicate that *ginkgo biloba* extract has some beneficial effects in the treatment of AD. In the clinical trials, we discovered that patients over 70 showed a low efficacy of GBE in AD treatment, while younger AD patients had effectively improved cognition after GBE treatment (Figure 3B). We speculate that, in the late stages of AD, the brain has generated irreversible lesions, making it difficult to alleviate the condition [66]. This also suggests that early diagnosis and intervention are imperative in order to delay the occurrence and development of AD. Shen et al. [67] found that the detection of β-secretase (BACE1) activity in the blood may predict the onset and progression of AD in the early clinical stage of mild cognitive impairment (MCI). Applying this insight to clinics can reduce the medical burden on families and society. In addition, a common flaw in the clinical trials is the lack of AD hallmark evaluations. Future clinical trials on GBE would be more convincing if the AD hallmarks were included as the key indicators of disease progression.

### 4.3. Future Perspective of GBE

#### 4.3.1. Nanomedicine Application

GBE is normally administered over a relatively long period (a few months) in animal models. Long-term (6 months), but not short-term (less than 6 months), GBE administration showed partial improvements in several AD clinical trials. A possible explanation is that GBE cannot cross the blood–brain barrier with high efficiency, which thereby limits its efficacy in improving cognition [68]. To solve this problem, Wang et al. [69] designed ginkgo- and corn-starch-based nanocarriers, according to the biocompatibility between ginkgo biloba extracts and starch, and they then loaded the GBE onto starch nanospheres (SNPs). Han et al. [70] designed a new system to achieve a synchronized and continuous release of EGB on the basis of an mPEG–PLGA–mPEG (PELGE) platform. Enhancing the brain penetration of GBE by pharmaceutical technique may be a feasible way to improve the efficacy of GBE.

#### 4.3.2. Drug Combination

The combination treatment of GBE and donepezil significantly decreased choline levels in aged rats [71], which were consistent with the clinical trial results [37]. Canevelli et al. [41] found that GBE provides some additional cognition improvements in AD patients already under ChEIs treatment. These data indicate that the combination of GBE with classic anti-AD drugs, including ChEIs, maybe be a way to improve AD treatment efficacy.

#### 4.3.3. Application in Other Neurodegenerative Diseases

Previous clinical trials have identified that the long-term administration of GBE is safe at doses of up to 240 mg/day [8,39,72,73]. Multiple preclinical studies and clinical trials also reveal the neuroprotective potential of GBE. It is reasonable to speculate that GBE may also exert therapeutic activity on other neurodegenerative diseases, including Parkinson’s disease. Indeed, several preclinical studies have revealed the potential protective effect of GBE on experimental PD models [74,75,76,77]. The neuroprotective activity of GBE, and the underlying mechanisms, deserve extensive investigation.

#### 4.3.4. Comparison of GBE Effects in Rodents and in Humans

By comparing the meta-analysis data from human clinical trials and rodent models, we can see that GBE generally displayed more consistent and striking activity in the rodent models. In the Morris water maze experiment, GBE significantly increased the numbers of times the animals crossed the target quadrant and decreased the escape latencies, when compared with vehicle-treated animals (*p* < 0.00001), for both mice and rats. However, the SKT and ADAS-Cog scores only decreased to a certain degree after GBE treatment, with the SMD (95%CI) = −2.44 [−2.60, −2.29] (score change of SKT), and the SMD (95%CI) = −0.57 [−0.68, −0.46] (score change of ADAS-Cog). We propose several possible reasons for this difference. First, rodents have much shorter life cycles than humans, and the duration of the GBE administration takes a larger proportion of the life cycles of mice than of humans. Second, the normalized GBE administration concentration is much higher in rodents than in humans. Third, experimental mice are inbred animals with homogenized genetic backgrounds, and the individual differences are much smaller in rodents than in humans. Fourth, the cognitive function measurements in rodent models are objective experiments, whereas, in humans, they are assessed on more subjective assessment scales.

### 4.4. Conclusions

Through the meta-analysis of preclinical studies, we find that GBE displayed predominantly positive anti-AD properties in animal models, by multiple mechanisms. Our analyses also suggest that a high dose (240 mg/day) and a prolonged (over 24 weeks) administration of GBE in the early stage of AD may support improved cognitive function. However, these results should be viewed with caution given the noted methodological concerns with regard to the reviewed publications. Considering the consistent safety determinations of long-term GBE administration, future clinical trials focusing on early-stage AD patients, or on a healthy aging population with long-term GBE administration (over 24 weeks) at a high dosage (>240 mg/day), may be helpful in determining the efficacy of GBE in the alleviation or prevention of AD.

## Figures and Tables

**Figure 1 cells-11-00479-f001:**
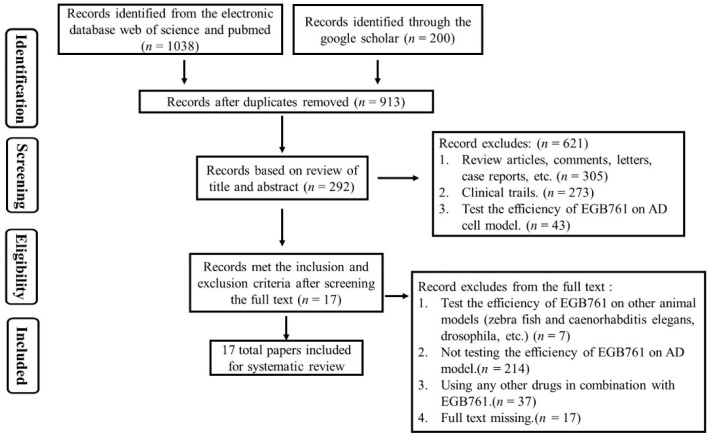
The screening flow chart of the preclinical studies.

**Figure 2 cells-11-00479-f002:**
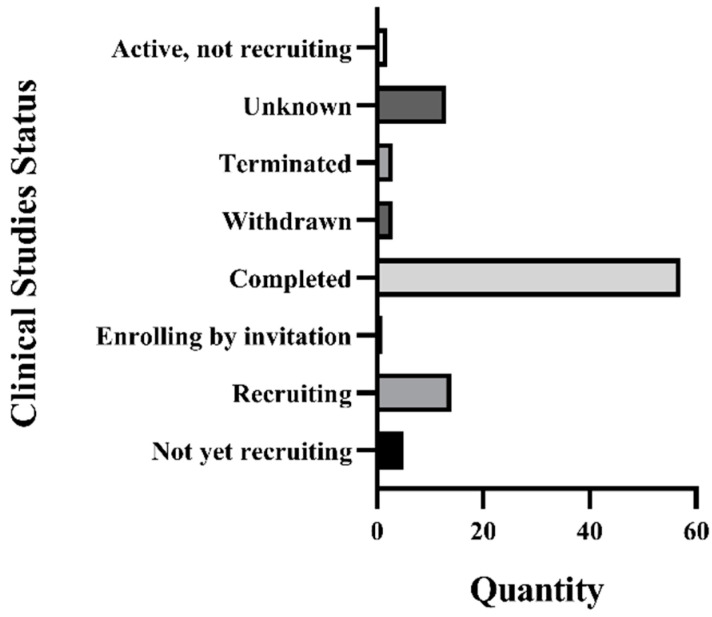
Recruitment status of GBE clinical trials.

**Figure 3 cells-11-00479-f003:**
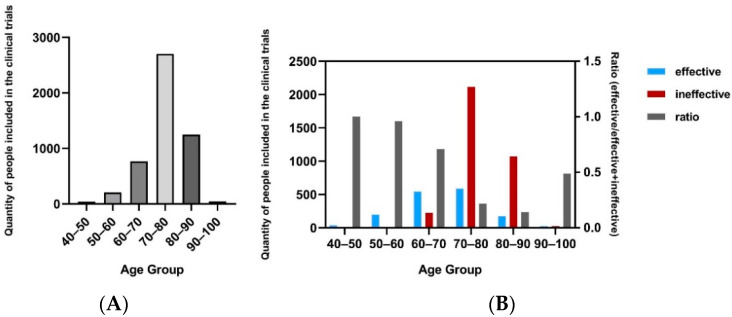
(**A**) The age distribution of all participants in clinical trials. (**B**) Comparison of age groups in clinical trial studies.

**Figure 4 cells-11-00479-f004:**
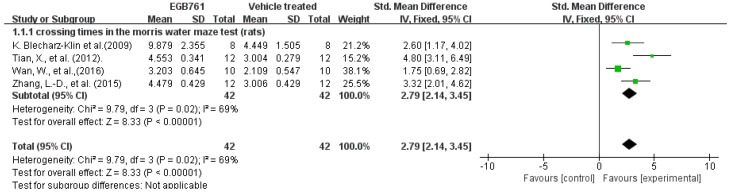
Forest plot for comparison: GBE treatment versus vehicle treatment. Outcome: numbers of times crossing the target quadrant in MWM. The square area represents the weight assigned to the study in the meta-analysis. The horizontal line represents the 95% confidence interval.

**Figure 5 cells-11-00479-f005:**
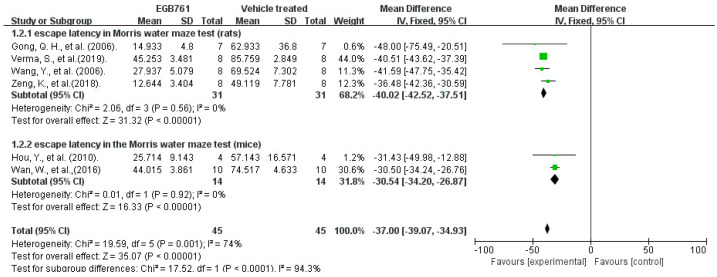
Forest plot for comparison: GBE treatment versus vehicle treatment. Outcome: escape latency in the probe test of MWM. The square area represents the weight assigned to the study in the meta-analysis. The horizontal line represents the 95% confidence interval.

**Figure 6 cells-11-00479-f006:**
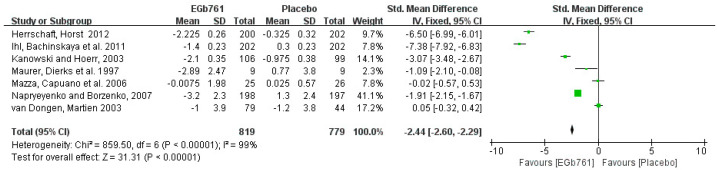
Meta-analysis of all patients in SKT. The square area represents the weight assigned to the study in the meta-analysis. The horizontal line represents the 95% confidence interval.

**Figure 7 cells-11-00479-f007:**
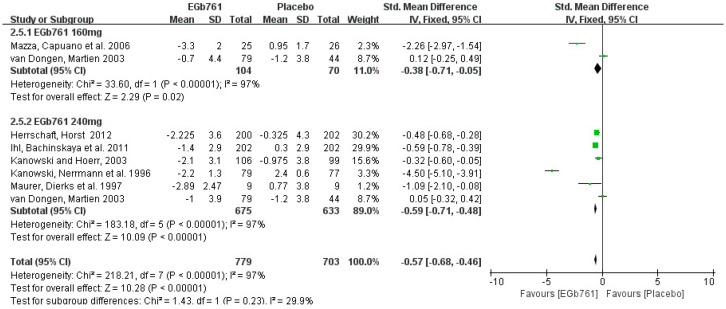
Forest plot of comparison: meta-analysis of AD patients taking either placebos or GBE in SKT, at doses of 240 mg/d, and below 240 mg/d. The square area represents the weight assigned to the study in the meta-analysis. The horizontal line represents the 95% confidence interval.

**Figure 8 cells-11-00479-f008:**
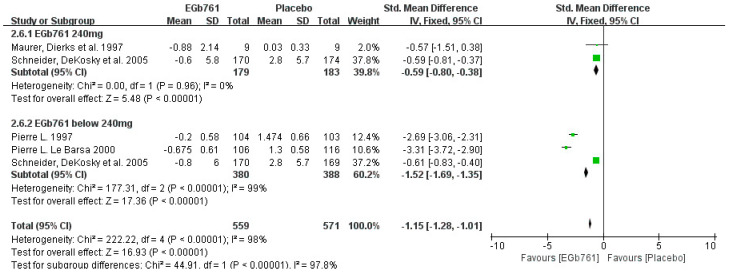
Forest plot of comparison: meta-analysis of ADAS-Cog scores in AD patients taking placebos or GBE, at doses of 240 mg/d, and below 240 mg/d. The square area represents the weight assigned to the study in the meta-analysis. The horizontal line represents the 95% confidence interval.

**Figure 9 cells-11-00479-f009:**
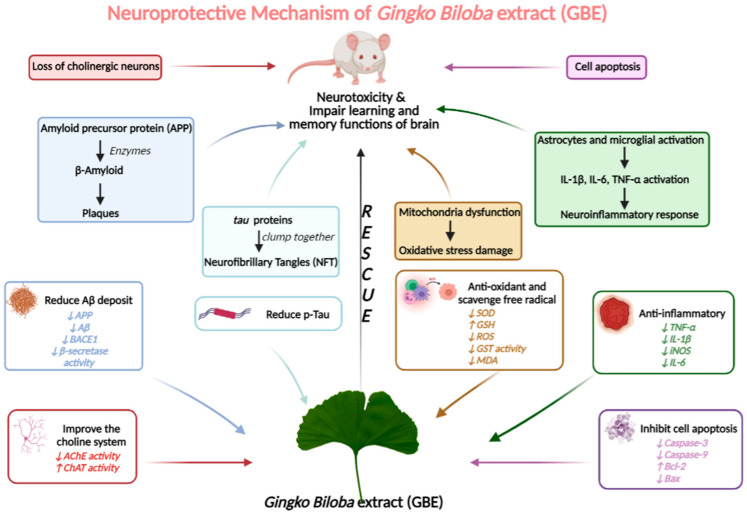
Possible neuroprotective mechanism of GBE.

**Figure 10 cells-11-00479-f010:**
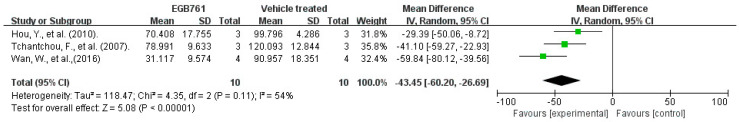
Forest plot for comparison: GBE treatment versus vehicle treatment. Outcome: Aβ density in the brain. The square area represents the weight assigned to the study in the meta-analysis. The horizontal line represents the 95% confidence interval.

**Table 1 cells-11-00479-t001:** Inclusion and exclusion criteria for selecting preclinical articles.

**Inclusion Criteria:**
1. Parallel experiments were conducted to evaluate the effects of EGB761 on AD protection in vivo.
2. Laboratory animals of any species, age, sex, or strain to induce AD models were included.
3. Any kind of EGB761 intervention compared with a control group was included. Dosages, methods of treatment, and curative times were not limited.
**Exclusion Criteria:**
Duplicated references; articles with incorrect and incomplete data; no access to the databases; review articles, comments, letters, and case reports.

**Table 2 cells-11-00479-t002:** Inclusion and exclusion criteria for selecting clinical articles.

**Inclusion Criteria:**
1. The clinical trials were designed as double-blind randomized placebo-controlled trials.
2. The patients, by age, sex, administration route and duration, dosage, were included in the trials.
3. Specific and reliable criteria for the AD assessment, such as the SKT and MMSE, were included.
**Exclusion Criteria:**
Duplicated references; repetitive clinical data; articles with incorrect and incomplete data; no access to the databases; review articles, comments, letters, and case reports.

**Table 3 cells-11-00479-t003:** Characteristics of included animal studies related to GBE.

Author	Animal Model	Treatment	Method	Result	Mechanism of Action
Ward, C. P. et al. (2002) [12]	C57BL/6 mice, male, 20 months old	EGb 761 (100 mg/kg/day), orally for 82 consecutive days	1. Morris water maze test	↑the time of hidden platform ↓platform crossings (probe test)	Improved the learning and memory cognition
			2. Elevated plus-maze test	↓time on the open arms	
			3. Protein levels of CREB	no significant differences	Antioxidant properties
Stackman, R. W. et al. (2003) [14]	Tg2576 mice, female, 8 months old	EGb 761 (70 mg/kg/day), orally for 6 months	1. Morris water maze test	↓average distance to the platform ↑search ratio	Alleviated the spatial learning impairment
			2. Fibrillar and soluble β-amyloid and protein oxidation products (ELISA)	↓soluble β-amyloid	N/A
			3. Histological assessment	↓β-amyloid	
			4. Protein carbonyl	↑protein carbonyls	Antioxidant properties
Gong, Q. H. et al. (2005) [15]	Wistar rats, male, 8–12 weeks old, daily AlCl_3_ solution, (500 mg/kg, i.g, 0.5 mL/100 g), gavage for 1 month	EGb761 (50 mg/kg/day, 100 mg/kg/day, 200 mg/kg/day), orally for 2 months	1. Morris water maze test	↓searching distance ↓escape latency	Ameliorated the learning and memory abilities
			2. Level of caspase-3	↓caspase-3	Antiapoptosis
			3. Level of APP (immunohistochemistry)	↓APP	N/A
Wang, Y. et al. (2006) [16]	1. Wistar rats, male, 12–13 weeks old 2. Wistar rats, male, 74–78 weeks old (aged)	1. EGb761 (30 mg/kg/day) 2. EGb761 (60 mg/kg/day), orally for 30 consecutive days	1. Morris water maze test	↓escape latency ↑search time	Improved spatial learning in aged animals
			2. Changes in synaptic plasticity	↑hippocampal LTP	N/A
Gong, Q. H. et al. (2006) [17]	Wistar rats, male, 8–12 weeks old, daily 50 g/L AICI_3_, gavage for 2 months	EGb761 (50 mg/kg/day, 100 mg/kg/day, 200 mg/kg/day), orally for 2 months	1. Morris water maze test	↓escape latency ↓searching distance	Reduced learning and memory deficits
			2. Effect of AChE activity	↓AChE activity	Cholinergic improvement
Tchantchou, F. et al. (2007) [18]	1. TgAPP/PS1 founder mice, 6 months old 2. TgAPP/PS1 founder mice, 22 months old	EGb761 (100 mg/kg/day), orally for 1 month	1. Determine the neurogenic potential	↑cell proliferation in the hippocampus	Induced neurogenesis as compensation
			2. Levels of Aβ and CREB/pCREB	↓Aβ oligomers ↑pCREB levels in the hippocampus	N/A
Blecharz-Klin, K. et al. (2009) [19]	Wistar rats, male 18 months old	1. EGb761 (50 mg/kg b.w./day); 2. EGb761 (100 mg/kg b.w./day); 3. EGb761 (150 mg/kg b.w./day); orally for 3 months	1. Morris water maze test	↓crossings ↓escape latency ↓mean swimming speed	Improved spatial memory
			2. Hole-board test	↑motor activity	
			3. HPLC detects the levels of DA, 5-HT, NA, and HVA	↑NA in prefrontal cortex and hippocampus ↓DA in prefrontal cortex and hippocampus ↑DOPAC in the prefrontal cortex ↓DOPAC in hippocampus ↑5-HT in the striatum	Neurotransmitter balance regulation
Hou, Y. et al. (2010) [20]	TgAPP/PS1 mice, male, 8 months old	1. Ginkgo biloba extract (50 mg/kg/day),gavage for 4 months; 2. flavonol (50 mg/kg/day), i.p. for 7 days	1. Morris water maze test	↓time needed to find the platform	Improved impaired spatial learning
			2. Levels of BDNF, pCREB, and Aβ	↑BDNF in neurons and hippocampus ↓both intracellular and medium Aβ levels	NMDA receptor Antagonist Anti-inflammatory activity
			3. Immunohistochemistry of Aβ deposition	↓Aβ deposition and plaque formation in hippocampus	N/A
Tian, X. et al. (2012) [21]	Sprague–Dawley rats, male, 3–4 months old, Aβ25-35 (1 µg/µL), i.c.v.	EGb761 (40 mg/tablets), gavage for 20 days	1. Morris water maze test	↓escape latencies ↑platform crossing times ↑percentage of swimming time in Quadrant 1	Improved the learning and memory cognition
			2. Histopathological changes in Aβ	↓density of the damaged neurons ↑neuronal number	Anti-inflammatory activity
			3. Activity of SOD, MDA, and NO	↓SOD ↓MDA ↓NO	Antioxidant properties
Tian, X. et al. (2013) [22]	Sprague–Dawley rats, male, 4–5 months old, Aβ25–35 (1 µg/µL), i.c.v.	EGB761 (20 mg/kg/day), gavage for 20 days	1. Morris water maze test	↓escape latency ↑platform crossings	Improved the learning and memory cognition
			2. Levels of SOD, GSH, and MDA	↓SOD ↑GSH ↓MDA	Antioxidant properties
			3. Levels of caspase-9 and caspase-3	↓caspase-9 ↓caspase-3	Inhibited cell apoptosis
			4. TUNEL staining	↓neuronal apoptosis	
			5. RT-PCR of Bcl-2	and Bax ↑Bcl-2 ↓Bax	Inhibited cell apoptosis
Jahanshahi, M. et al. (2013) [23]	Wistar rats, male, Scopolamine (3 mg/kg), intraperitoneal injection	Ginkgo biloba extract (40 and 80 mg/kg, IP), everyday injection for a week	1. TUNEL staining	↓apoptotic cells in the hippocampus	Antioxidant and hydroxyl radical scavenging activity
Zhang, L.-D. et al. (2015) [24]	Sprague–Dawley rats, male, 5–6 months old, Aβ25–35 (10 μL; 1 g/L), i.c.v.	EGB761 (20 mg/kg/day), gavage for 20 days	1. Morris water maze test	↑times of crossing the former platform ↑percentage of time spent in the quadrant	Improved cognitive and memory capacities
			2. TUNEL staining	↓brown precipitate (apoptosis identification)	Inhibited cell apoptosis
			3. Levels of p-IKKα/β, p-IκBα, and p-NFκB	↑p-IKKα/β ↑p-IκBα ↑p-NFκB	Anti-inflammatory activity
Liu, X. et al. (2015) [25]	TgCRND8 APP-transgenic mice, female, 2 months old	EGb761 (600 mg/kg/day) (0.6%), orally for 5 months	1. Barnes maze test	↓time and ↓distance to reach the escape chamber	Improved cognitive function
			2. Level of Aβ (ELISA) (%)	↓Aβ	N/A
			3. Immunofluorescent staining of Aβ		
			4. Histological analysis of Iba1	↓Iba1 positive cell number	Neuroinflammatory inhibition
			5. Levels of tnf-α, il-1β, ccl-2, and IL-10	↓TNF-α, IL-1β, ccl-2, iNOS, and IL-10	
Wan, W. et al. (2016) [26]	APP/PS1 transgenic mice, male, 2 months old	EGb761 (50 mg/kg/day), orally for 6 months	1. Morris water maze test	↓escape latency ↓time of passing the platform ↑crossing times	Improved cognitive function
			2. Level of Aβ (ELISA)	↓Aβ	N/A
			3. Ratio of fluorescence intensity	↑microglia around the plaque	Attenuated inflammatory reactions
Zeng, K. et al. (2018) [27]	Sprague–Dawley rats, male, 8 weeks old, Hhcy (400 μg/kg/day), for 14 days i.p.	EGb761 (400 mg/kg/day), gavage for 7 days	1. Morris water maze test	↓escape latency	Ameliorated memory deficits
			2. Levels of SOD and MDA	↓SOD↓MDA	Antioxidant properties
			3. Levels of tau phosphorylation, PSD95, and synapsin-1	↓tau phosphorylation ↑PSD95 ↑synapsin-1	Attenuated oxidative damage
Verma, S. et al. (2019) [28]	Sprague–Dawley rats, female, 12 months old, Al(lac)_3_ (10 mg/kg b.wt), daily for 6 weeks	Ginkgo biloba extract, EGb761 (100 mg/kg/day), orally for 6 weeks	1. Morris water maze test	↓time to find the platform ↓escape latency	Improved spatial memory
			2. Histopathological changes in Aβ	↓ThT positive cells in hippocampusand cortex ↓Congo red	Antioxidative stress
			3. Levels of 5-HT, GSH, GST, and SOD	↑5-HT↓SOD↑GSH↓GST	
			4. AChE activity	↓AChE activity in the hippocampus and cortex	
Verma, S. et al. (2020) [29]	Sprague–Dawley rats, female, 12 months old, Al(lac)_3_ (10 mg/kg b.wt), daily for 6 weeks	Ginkgo biloba extract, EGb761 (100 mg/kg/day), orally for 6 weeks	1. Morris water maze test	↓escape latency	Prevented behavioral impairments
			2. Level of ROS	↓ROS	Antioxidative stress
			3. Protein level of APP, Aβ, and p-Tau (ELISA)	↓APP ↓Aβ ↓p-Tau	N/A
			4. Histopathological changes	↓silver positive deposits in CA1, CA3 ↓congo red positive deposits in CA1, CA3 ↓ThT positive deposits	Antioxidative stress
			5. AchE activity	↓AChE enzyme activity	Cholinergic improvement Neurotransmitter balance regulation
			6. Level of MAO-B	↓MAO-B enzyme activity	
			7. Immunohistochemistry of Aβ (17–23)	↓Aβ (17–23)	N/A

**Table 4 cells-11-00479-t004:** Characteristics of included clinical studies related to GBE.

Study Author, Date	Country	Inclusion Criteria	Setting	Duration	Treatment	Groups	Age	Baseline Scale				Withdrawal Rate
								Cognition		Age	Female (%)	
Effective												
Schaffler and Reeh, 1985 [30]	United Kingdom	/	Normal healthy volunteers	2 weeks	EGB (Tebonin) 80 mg/day	EGB: *n* = 4 Placebo: *n* = 4	27	/	/	27.3 ± 2.6	0	/
Wesnes et al., 1987 [31]	United Kingdom	Crichton geriatric behavioral scale > 14	Outpatient	12 weeks	EGB (Tanakan) 120 mg/day	EGB: *n* = 27 Placebo: *n* = 27	62~85	/	/	70.7 ± 7.1 71.3 ± 6.6	30% 44%	7%
Rai et al., 1991 [32]	United Kingdom	NINCDS-ADRDA diagnostic criteria	Outpatient	6 months	EGB (Tanakan) 120 mg/day	EGB: *n* = 12 Placebo: *n* = 15	>50	MMSE	26.8 24.3	73.4 ± 7.3 78.3 ± 5.9	67% 80%	13%
								Kendrick digit copying task	106.6 94.53			
								Kendrick object learning task	93.17 87.27			
Kanowski, Nerrmann et al., 1996 [33]	Germany	SKT: 6~18; MMSE: 13~25	Outpatient	24 weeks	EGb761 240 mg/day	EGb761: *n* = 79 Placebo: *n* = 77	>55	SKT	10.2 ± 3.0 11.2 ± 3.4	70 ± 10 68 ± 10	66% 69%	30%
Maurer, Dierks et al., 1997 [34]	Germany	DSM-III-R and ICD-10 criteria; Hachinski ischemic score < 4 mean; BCRS score 3–5	Outpatient	12 weeks	EGb761 240 mg/day	EGb761: *n* = 10 Placebo: *n* = 10	50~80	SKT	19.7 ± 6.4 18.1 ± 9.4	68.5 ± 6 60.6 ± 8.2	56% 45%	10%
								ADAS-Cog	31.2 ± 12.6 36.1 ± 15.2			
Barsa, Kieserc et al., 2000 [35]	United States	DSM-III-R and ICD-10 criteria; MMSE: 9~26; global deterioration scale: 3~6	Outpatient	26 weeks	EGb761 120 mg/day	EGb761: *n* = 166 Placebo: *n* = 161	>45	MMSE	21.1 ± 5.8 21.2 ± 5.5	69 ± 10 69 ± 10	51% 56%	21%
								ADAS-Cog	20.0 ± 16.0 20.5 ± 14.7			
Kanowski and Hoerr, 2003 [36]	Germany	DSM-III-R and ICD-10 criteria; SKT: 6~18; MMSE: 13~25	Outpatient	24 weeks	EGb761 240 mg/day	EGb761: *n* = 106 Placebo: *n* = 99	>55	MMSE	21.6 ± 2.6 21.5 ± 2.4	72 ± 10 72 ± 10	68% 71%	7.65%
								SKT	10.5 ± 3.2 11.2 ± 3.3			
								ADAS-Cog	19.0 ± 4.1 19.9 ± 4.3			
Mazza, Capuano et al., 2006 [37]	Italy	Brief cognitive rating scale: 3~5; Hachinski ischemic score < 4; SKT: 8~23; MMSE: 13~25	Outpatient	24 weeks	EGb761 160 mg/day	EGb761: *n* = 25 donepezil: *n* = 25 Placebo: *n* = 16	50~80	MMSE	18.8 ± 3.6 18.8 ± 3.6	66.2 ± 6 64.5 ± 6 69.8 ± 3	52% 48% 61%	19.70%
								SKT	16.5 ± 3.1 15.9 ± 3.9			
Napryeyenko and Borzenko, 2007 [38]	Ukraine	NINCDS/ADRDA diagnostic criteria: SKT: 9~23; MMSE: 14~25; ADAS-Cog: 17~35	Outpatient	22 weeks	EGb761 240 mg/day	EGb761: *n* = 198 Placebo: *n* = 197	>50	SKT	15.6 ± 3.9 15.4 ± 3.7	65 ± 8 63 ± 8	72% 72%	1.25%
Ihl, Bachinskaya et al., 2011 [39]	Ukraine	NINCDS-ADRDA criteria; SKT: 9~23; MMSE: 14~25; ADAS-Cog: 17~35	Outpatient	24 weeks	EGb761 240 mg/day	EGb761: *n* = 206 Placebo: *n* = 204	>50	SKT	16.7 ± 3.9 17.2 ± 3.7	65 ± 10 65 ± 9	69% 66%	6.82%
Herrschaft, Nacu et al., 2012 [8]	Republic of Belarus, Republic of Moldova, and Russian Federation	NINCDS-ADRDA criteria; NINDSAIREN criteria; NINDS-AIREN crtteria	Outpatients	24 weeks	EGb761 240 mg/day	EGb761: *n* = 206 Placebo: *n* = 204	>50	SKT	15.1 ± 4.1 15.3 ± 4.2	65.1 ± 8.8 64.9 ± 9.4	69.5% 69.3%	2.00%
								NPI	16.8 ± 6.9 16.7 ± 6.4			
Amieva, Meillon et al., 2013 [40]	France	/	Outpatient	20 years	EGb761 dosage unclear	EGb761: *n* = 589 Piracetam: *n* = 149 Placebo: *n* = 2874	>65	MMSE	26.3 ± 2.9 25.7 ± 3.9 25.7 ± 3.5	74.8 ± 6.6 75.7 ± 6.6 75 ± 6.9	73.9% 61.1% 54.1%	0
Canevelli, Adali et al., 2014 [41]	Europe	NINCDS-ADRDA criteria, MMSE: 10~26	Outpatients	1 year	EGb761 120 mg/day	EGb761 + ChEIs: *n* = 29 ChEIs: *n* = 799	68~84	MMSE	21.2 ± 3.5 20.5 ± 3.9	76.2 ± 6.87 5.8 ± 7.8	62.1% 64.8%	0
								ADAS-Cog	15.8 ± 7.9 20.6 ± 8.9			
Hoerr and Nacu, 2016 [42]	Russian Federation, Republic of Belarus, Republic of Moldova	SKT: 9~23, mild to moderate dementia; test for the early detection of dementia with differentiation from depression ≤ 35	Outpatient	24 weeks	EGb761 240 mg/day	EGb761: *n* = 200 Placebo: *n* = 202	>65	SKT	15.1 ± 4.1 15.3 ± 4.2	65.1 ± 8.8 64.9 ± 9.4	69.5% 69.3%	2%
Ineffective												
Subhan and Hindmarch, 1984 [43]	United Kingdom	/	Normal healthy volunteers	1 h	EGb 761 120 mg/240 mg/600 mg	EGb761(120): *n* = 2 EGb761(240): *n* = 2 EGb761(600): *n* = 2 Placebo: *n* = 2	32	/	/	32 ± 0	100%	/
Schneider, DeKosky et al., 2005 [44]	United States	NINCDS/ ADRDA criteria; modified Hachinski ischemic score < 4; MMSE: 10~24	Outpatients	26 weeks	EGb761 120/240 mg/day	EGb761(120): *n* = 169 EGb761(240): *n* = 170 Placebo: *n* = 174	>60	MMSE	17.4 ± 3.8 (240) 17.9 ± 4.5 (120) 17.6 ± 3.9	78.6 ± 7.0 78.1 ± 7.0 77.5 ± 7.4	50% 56% 52%	20.00%
								ADAS-Cog	24.8 ± 11.3 (240) 26.8 ± 13.7 (120) 26.2 ± 11.8			
McCarney, Fisher et al., 2008 [45]	United Kingdom	DSM-IV criteria; MMSE: 12~26	Outpatient	24 weeks	EGb761 120 mg/day	EGb761: *n* = 88 Placebo: *n* = 88	>55	MMSE	23 22	79.3 ± 7.7 79.7 ± 7.5	58.0% 63.6%	25.60%
								ADAS-Cog	20.4 ± 8.2 25 ± 10.3			
Snitz, O’Meara et al., 2009 [46]	United States	MMSE; ADAS-Cog; neuropsychological test	community-dwelling participants	6.1 years	EGb761 240 mg/day	EGb761: *n* = 1545 Placebo: *n* = 1524	72~96	MMSE	93.4 ± 4.7 93.3 ± 4.7	79.1 ± 3.3 79.1 ± 3.3	45% 47%	37.80%
								ADAS-Cog	6.5 ± 2.86.4 ± 2.7			
Vellas, Coley et al., 2012 [47]	France	MMSE: >25; covianxiety scale <6; geriatric depression scale <15	Outpatient	5 years	EGb761 240 mg/day	EGb761: *n* = 1419 Placebo: *n* = 1435	>70	MMSE	27.6 ± 1.9 27.6 ± 1.9	76.3 ± 4.4 76.3 ± 4.4	67% 66%	31%
Nasab, Bahrammi et al., 2012 [48]	Iran	DSM IV criteria; NINCDS-ADRDA criteria; MMSE: 10~24	Outpatients	24 weeks	EGb761 120 mg/day	EGb761: *n* = 25 Rivastigmine: *n* = 25	50–75	MMSE	15.6 ± 4.1 16.6 ± 4.0	65.7 ± 4.7 66.0 ± 4.6	52% 57.7%	9.00%

**Table 5 cells-11-00479-t005:** Methodological qualities of GBE animal studies.

Methodological Quality Scores of Included Preclinical Studies									
	1	2	3	4	5	6	7	8	Total
Ward, C. P. et al. (2002) [12]	✔	✔			✔		✔		4
Stackman, R. W. et al. (2003) [14]	✔		✔		✔		✔		4
Gong, Q. H. et al. (2005) [15]	✔	✔		✔	✔		✔		5
Wang, Y. et al. (2006) [16]	✔	✔		✔	✔		✔		5
Gong, Q. H. et al. (2006) [17]	✔	✔		✔	✔		✔		5
Tchantchou, F. et al. (2007) [18]	✔				✔		✔		3
Blecharz-Klin, K. et al. (2009) [19]	✔	✔			✔		✔	✔	5
Hou, Y. et al. (2010) [50]	✔				✔		✔		3
Tian, X. et al. (2012) [21]	✔	✔			✔		✔	✔	5
Tian, X. et al. (2013) [22]	✔	✔			✔		✔		4
Jahanshahi, M. et al. (2013) [23]	✔	✔			✔		✔	✔	5
Zhang, L.-D. et al. (2015) [24]	✔	✔			✔		✔	✔	5
Liu, X. et al. (2015) [25]	✔	✔			✔		✔		3
Wan, W. et al. (2016) [26]	✔				✔		✔	✔	4
Zeng, K. et al. (2018) [27]	✔				✔		✔	✔	4
Verma, S. et al. (2019) [28]	✔				✔		✔	✔	4
Verma, S. et al. (2020) [29]	✔	✔			✔		✔	✔	5
1.Was the article published in a peer-reviewed journal?									
2. Were the animals allocated to the treatment group or control group randomly during the experiment?									
3. Were the outcomes assessed blindly?									
4. Was the dose–response relationship assessed during the experiment?									
5. Was the appropriate animal model used in the experiment?									
6. Was the necessary sample size calculated to achieve sufficient power?									
7. Were the animal welfare regulations complied with during the experiment?									
8. Was the study free of any potential conflicts of interest?									

**Table 6 cells-11-00479-t006:** Methodological quality scores of GBE-related clinical trials.

Methodological Quality Scores of Clinical Studies										
	1	2	3	4	5	6	7	8	9	Total
Schaffler and Reeh, 1985 [30]	✔	✔	✔		✔					4
Wesnes et al., 1987 [31]	✔	✔	✔		✔					4
Rai et al., 1991 [32]	✔	✔	✔		✔					4
Kanowski, Herrmann et al., 1996 [33]	✔	✔	✔		✔			✔	✔	6
Maurer, Dierks et al., 1997 [34]	✔	✔	✔		✔					4
Barsa, Kieserc et al., 2000 [35]	✔	✔	✔		✔	✔	✔	✔		7
Kanowski and Hoerr, 2003 [36]	✔	✔	✔				✔			5
Mazza, Capuano et al., 2006 [37]	✔	✔	✔							3
Napryeyenko and Borzenko, 2007 [38]	✔	✔	✔		✔	✔	✔	✔		7
Ihl, Bachinskaya et al., 2011 [39]	✔	✔	✔		✔		✔	✔	✔	7
Herrschaft, Nacu et al., 2012 [8]	✔	✔	✔		✔			✔	✔	6
Amieva, Meillon et al., 2013 [40]	✔					✔		✔		3
Canevelli, Adali et al., 2014 [41]	✔							✔	✔	3
Hoerr and Nacu, 2016 [42]	✔	✔	✔		✔	✔		✔	✔	7
Subhan and Hindmarch, 1984 [43]	✔	✔	✔		✔					4
Schneider, DeKosky et al., 2005 [44]	✔	✔	✔	✔	✔		✔	✔		7
McCarney, Fisher et al., 2008 [45]	✔	✔	✔		✔		✔	✔	✔	7
Snitz, O’Meara et al., 2009 [46]	✔	✔	✔		✔	✔		✔		5
Vellas, Coley et al., 2012 [47]	✔	✔	✔		✔	✔		✔	✔	7
Nasab, Bahrammi et al., 2012 [48]	✔	✔	✔		✔					4
1.Was the article published in a peer-reviewed journal?										
2.Were the patients allocated randomly during the clinical trial?										
3. Were the outcomes assessed blindly?										
4. Was the dose–response relationship assessed during the clinical trial?										
5. Were the withdrawals per group reported during the clinical trial?										
6. Was the necessary sample size calculated to achieve sufficient power?										
7. Was the ITT analysis (intent-to-treat analysis) conducted?										
8. Was the funding reported for the clinical trial?										
9. Was the study free of potential conflicts of interest?										

## Data Availability

Not applicable.

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
