# Peer review of "Can We Use Ginkgo biloba Extract to Treat Alzheimer’s Disease? Lessons from Preclinical and Clinical Studies"

_cells, 2022, doi:10.3390/cells11030479_

Round 1

Reviewer 1 Report

See attached file

Reviewer 2 Report

In this systematic review and meta-analysis the authors have reported the results of  systematic meta-analysis of articles reporting pre-clinical and clinical studies of EGB in the treatment of AD. In addition, they summarized the neuroprotective mechanisms of EGB in AD as determined from animal models. However, before publication can take place, there are several concerns to attend to:

1. Major revision:

– Why did the authors decide to accomplish their research using only PubMed and Web of Science to find randomized controlled trials? From my personal point of view, other different online databases (e.g, Scopus, Google scholar, Biomed Central, Cochrane, Embase etc.) should be used to improve the impact of this systematic review.

– The authors should provide details on studies' inclusion/selection in order to check, during the review process, if they have appropriately performed the studies selection. On which basis did the authors eliminate those manuscripts deemed irrelevant?

– How could the authors justify that heterogeneity between studies, if the latter exists?

– The whole manuscript is also characterized by a series of imprecisions and inaccuracies.

- Try to separate clearly in your review article clinical results from experimental studies (i.e. studies on cells, in vitro, animals)

– The figures 4-9 shouldbe displayed as tables instead of fugures. However, if the authors will decide to show them as figures, the resolution should be improved.

The tables should be increased in size and of high quality resolution.

- EGB should be replaced by GBE throughout the manuscript.

Reviewer 3 Report

Ginkgo biloba Extract has been well demonstrated as a promising medicine to fight against neurodegenerative diseases, especially Alzheimer’s disease. In this review, the authors provided a comprehensive and updated discussion on the effectiveness of Ginkgo biloba Extract in AD treatment using both pre-clinical and clinical studies. The paper is well written and will be very informative for the readers. I only have some minor suggestions for the authors:

  1. It will be more informative if the authors can make a figure to summarize the neuroprotective mechanisms of Ginkgo biloba Extract, as discussed in the main text.
  2. Please provide a separate section to discuss the Future direction of EGB application in AD therapy, especially highlight the Lessons from Preclinical and Clinical Studies as shown in the title.

Reviewer 4 Report

The manuscript by Dr Xie et al., entitled “Can We Use Ginkgo biloba Extract to Treat Alzheimer’s Dis-2 ease? Lessons from Preclinical and Clinical Studies”, deals with a systematical evaluation of electronic databases concerning 17 pre-clinical studies and 20 clinical trials assessing the therapeutic effects of Ginkgo Biloba extract (EGB) against Alzheimer’s disease (AD). The Authors found that chronic administration of EGB displayed anti-AD effects in animal models and in early-stage AD patients, these latter evaluated as cognitive ability in SKT and ADAS-Cog scores.

The meta-analysis of database is carried out accurately, when referring to both pre-clinical and clinical studies. The selection of the studies agrees with the inclusion criteria chosen by the Authors, and their analysis shows a critical insight, which allows to further select among the selected studies those with the highest reliability and scientific soundness. This applies especially to pre-clinical studies, which are critically reviewed on the basis of their ability to better and fully reproduce the pathological features of AD.

The article is well written and it addresses the main item of the paper with methodological rigor and critical analysis of the data collected. Discussion is adequate.

I have only a question about pathological outcomes in humans. Are there any autopsy report about AD patients who participated to clinical trials with EGB? It is of interest to document the impact of EGB on the main AD hallmarks such as Abeta plaques and/or neurofibrillary tangles.

Round 2

Reviewer 1 Report

While the authors attempted to address the issues specifically mentioned in the review, these were examples of larger problems in the manuscript and I did not spell out every similar case. Therefore, similar problems still exist in the manuscript as the authors did not review their own manuscript for the comparable issues found elsewhere. Please apply the examples given throughout the manuscript.

The authors have not yet adequately discussed the problem of methodological issues in the reviewed papers. Noting that problems exist is not the same as weighing the validity of the results against the methodological errors. Additional, thoughtful discussion of the influence of these errors on the mega-analysis and adding a validity weighted measure to the analysis might help.

The authors have introduced new grammatical errors in their rewritten text which need to be addressed.

Figure 10 should be more cautious in the title, since the reviewed research does not support definitive conclusions: “Figure 10. Proposed Neuroprotective Mechanism of GBE”

Given the overwhelming problems in the reviewed paper, I suggest that the conclusions should be more cautiously stated than they currently are.

Suggested rewording:

“4.4 Conclusions

Through the meta-analysis of preclinical, we find that GBE displayed predominantly positive anti-AD properties in animal models by multiple mechanisms. Our analyses also suggests that a high dose (240mg/day) and prolonged (over 24 weeks) administration of GBE in the early stage of AD may support improved cognitive function. However, these results should be viewed with caution given the noted methodological concerns of the reviewed publications. Considering consistent safety determinations of long-term GBE administration, future clinical trials focusing on early-stage AD patients, or a healthy aging population with long-term GBE administration (over 24 weeks) at high dosage (>240mg/day) may be helpful in determining the efficacy of GBE in the alleviation or prevention of AD."

It would be helpful for the authors to explicitly state whether the effect of GBE across the reviewed papers was generally greater / equal / or smaller for mice studies than for the human clinical studies? This is an important consideration for all studies on AD which often reveal a stronger effect on rodents than that found in the human clinical studies.

Reviewer 2 Report

This manuscript shows significant improvements. I thank the authors for considering all my comments/suggestions.

Author Response

Dear Reviewer,

Thanks again for your nice comments!